# YTHDF1 Attenuates TBI-Induced Brain-Gut Axis Dysfunction in Mice

**DOI:** 10.3390/ijms24044240

**Published:** 2023-02-20

**Authors:** Peizan Huang, Min Liu, Jing Zhang, Xiang Zhong, Chunlong Zhong

**Affiliations:** 1Department of Neurosurgery, Shanghai East Hospital, Nanjing Medical University, Shanghai 200120, China; 2Department of Neurosurgery, Shanghai East Hospital, School of Medicine, Tongji University, Shanghai 200120, China; 3Department of Neurosurgery, The Fourth Affiliated Hospital of Nanjing Medical University, Nanjing 210031, China; 4College of Animal Science and Technology, Nanjing Agricultural University, Nanjing 210095, China

**Keywords:** YTHDF1, traumatic brain injury, brain-gut axis, microbiota, m^6^A RNA modification

## Abstract

The brain-gut axis (BGA) is a significant bidirectional communication pathway between the brain and gut. Traumatic brain injury (TBI) induced neurotoxicity and neuroinflammation can affect gut functions through BGA. N^6^-methyladenosine (m^6^A), as the most popular posttranscriptional modification of eukaryotic mRNA, has recently been identified as playing important roles in both the brain and gut. However, whether m^6^A RNA methylation modification is involved in TBI-induced BGA dysfunction is not clear. Here, we showed that YTHDF1 knockout reduced histopathological lesions and decreased the levels of apoptosis, inflammation, and oedema proteins in brain and gut tissues in mice after TBI. We also found that YTHDF1 knockout improved fungal mycobiome abundance and probiotic (particularly *Akkermansia*) colonization in mice at 3 days post-CCI. Then, we identified the differentially expressed genes (DEGs) in the cortex between YTHDF1-knockout and WT mice. These genes were primarily enriched in the regulation of neurotransmitter-related neuronal signalling pathways, inflammatory signalling pathways, and apoptotic signalling pathways. This study reveals that the *ITGA6*-mediated cell adhesion molecule signalling pathway may be the key feature of m^6^A regulation in TBI-induced BGA dysfunction. Our results suggest that YTHDF1 knockout could attenuate TBI-induced BGA dysfunction.

## 1. Introduction

Traumatic brain injury (TBI) is a major public health concern, with up to 69 million TBIs occurring worldwide each year [1,2]. Due to the poor understanding of TBI’s heterogeneity and complexity, there has been limited clinical success, leading to a huge social and economic burden. TBI exerts profound effects on the gut’s functions [3]. The brain–gut axis (BGA), a bidirectional communication network connecting the central nervous system (CNS) and enteric nervous system (ENS), is the key to CNS and gastrointestinal homeostasis and regulates diverse functions including gut barrier functions, intestinal motility, and neurobehaviours [4]. TBI-induced neurotoxicity and neuroinflammation can affect gut functions through BGA [5,6]. However, the mechanisms of BGA dysfunction induced by TBI are still elusive, especially in posttranscriptional regulation.

N6-methyladenosine (m^6^A), the most prevalent mRNA posttranscriptional modification in eukaryotes, requires various regulatory proteins encoded by writing genes (writers), erasing genes (erasers), and reading genes (readers) [7]. An increasing amount of research has revealed that m^6^A modification can influence almost all aspects of RNA metabolism, including RNA transcription, splicing, nuclear output, translation, decay, and RNA-protein interactions [8,9,10,11]. Substantial lines of study have shown that m^6^A exquisitely regulates various spatial and temporal physiological processes, including gametogenesis, embryogenesis, cell fate determination, sex determination, DNA damage response, circadian rhythm, heat shock response, pluripotency, and neuronal functions [10,12,13]. It has recently been identified that m^6^A plays significant roles in both the brain and gut. m^6^A modification is abundant in the CNS, modulates the activation of various nerve conduction pathways, and plays an important role in the development, differentiation, and regeneration of neurons [14]. Meanwhile, it shows a significant impact on the communication between the gut microbiome and the host [15,16]. Thus, m^6^A modification may also play an important role in BGA. As one of the m^6^A reader proteins, YTHDF1 promotes mRNA translation. The knockout of YTHDF1 does not alter the m^6^A/A ratio of total mRNA but impacts on the association of its target mRNA with ribosome [17]. YTHDF1 regulates the subcellular distribution and translation status of the m^6^A-modified mRNA [17]. YTHDF1 plays an important role in both the brain and gut. In the brain, YTHDF1 recognizes and binds the mRNAs of m^6^A-modified glutamate ionotropic receptor NMDA type subunit 1 and 2A (*GRIN1*, *GRIN2A*), glutamate ionotropic receptor AMPA 1 (*GRIA1*), calcium/calmodulin-dependent protein kinase (CaM kinase) II alpha and II beta (*CAMK2A*, *CAMK2B*) genes to promote translation, while YTHDF1 deletion causes the ectopic translation of related proteins, resulting in memory loss [18]; moreover, the upregulation of YTHDF1 promotes cancer stem cell properties in glioblastoma cells [19]. In the gut, YTHDF1 recognizes the target TNF receptor associated factor 6 (*TRAF6*) transcript to modulate the gut immune response to bacterial infection, by the unique interaction mechanism between the P/Q/N-rich domain and host factor DEAD (Asp-Glu-Ala-Asp) box polypeptide 60 (DDX60) death domain [20]. YTHDF1-mediated exportin 1 (XPO1) activates the NF-κB pathway and then induces an increased expression of IL-8 in gut cells, resulting in the development and aggravation of celiac disease [21]. YTHDF1 exhibits the highest diagnostic value for chronic obstructive pulmonary disease (COAD) [22]. However, the roles of YTHDF1 in the TBI-induced brain-gut axis dysfunction are unclear; therefore, the functions of YTHDF1 in TBI need to be investigated. To confirm the involvement of m^6^A RNA methylation in TBI-induced BGA dysfunction, we performed controlled cortical impact (CCI) on YTHDF1-knockout mice and C57BL/6J mice, and compared their differences in the brain defect area; surviving neuron count; the levels of apoptosis, inflammation, and oedema proteins in brain tissues; the ratio of villus height to crypt depth (V/C); the levels of apoptosis and inflammation proteins in gut tissues; and the composition of the faecal microbiome. Then, using RNA-seq, we identified the differentially expressed genes (DEGs) in the cortex and further explored the possible pathways and key genes of YTHDF1 regulating TBI-induced BGA dysfunction.

## 2. Results

### 2.1. YTHDF1-Knockout Decreases the Cortical Tissue Losses while Increasing the Neuronal Cell Survivals and the Colon Tissue V/C Ratio after CCI

Three days after CCI, HE revealed that there was no significant brain tissue loss in either the WT + sham group or the YTHDF1-knockout + sham group, while marked brain tissue losses were observed in the CCI groups compared with the sham groups (Figure 1B). Notably, compared with the YTHDF1-knockout + CCI group, there was a significant loss of cortical tissue in the WT + CCI group (Figure 1B,C) (*p* < 0.05). In addition, compared with the sham groups, the YTHDF1-knockout + CCI group and WT + CCI group displayed a significant decrease in the total neuron count in the perilesional zone to trauma, with a greater decrease in the WT + CCI group. However, neuronal cell survivals were not significantly different between the YTHDF1-knockout + sham group and the WT + sham group (Figure 1D,E).

In the colon tissue at 3 days after CCI, the villus height decreased while the crypt depth increased, and the V/C ratios of the YTHDF1-knockout + CCI group and WT + CCI group significantly decreased compared with that of the sham groups. Notably, the V/C ratio of the WT + CCI group decreased more markedly than that compared with the YTHDF1-knockout + CCI group. However, there was no significant difference between the YTHDF1-knockout + sham group and the WT + sham group (Figure 2).

### 2.2. YTHDF1-Knockout Decreases Pro-Apoptosis, Pro-Inflammation and Pro-Oedema Protein Levels and Increases the Anti-Apoptosis and Anti-Oedema Protein Levels following CCI

At 3 days after CCI, Western blots revealed that the apoptosis proteins BCL2-associated X protein (BAX), caspase3, and Cleaved caspase3 were notably decreased, while B-cell CLL/lymphoma 2 (BCL2) was markedly increased in the YTHDF1-knockout group compared with the WT group in injured cortical tissues (Figure 3A). Meanwhile, compared with the WT group, the inflammatory protein CD68 significantly decreased, forkhead box P3 (FOXP3) markedly increased (Figure 3B), and the oedema protein aquaporin 4 (AQP4) notably decreased, while claudin5 markedly increased (Figure 3C) in the YTHDF1-knockout group.

In colon tissues, the pro-apoptosis proteins, such as BAX, Caspase3, and Cleaved caspase3 in the YTHDF1-knockout group, displayed a significant decrease, while the anti-apoptosis protein BCL2 exhibited a notable increase compared with the WT group at 3 days post-CCI (Figure 4A). In addition, the pro-inflammation protein CD68 reduced markedly and the anti-inflammation protein FOXP3 was significantly enhanced in the YTHDF1-knockout group compared with the WT group (Figure 4B).

### 2.3. YTHDF1-Knockout Increases the Abundance of Fungal Mycobiome, and Alters Mycobiome Structure and Mycobiome Colonization after CCI

The raw data from the Illumina (San Diego, CA, USA) sequencing were processed using the Quantitative Insights Into Microbial Ecology (QIIME2) pipeline, and their relative abundance was calculated and grouped according to mouse origin (Figure 5). The results revealed the different profiles of fungal communities in the faecal mycobiomes of YTHDF1-knockout and WT mice at 3 days after CCI. The majority of fungal communities were in the *Firmicutes* phylum, which dominated the faecal mycobiome of mice, contributing approximately > 50.7% of the total identified fungi. The second major phylum in the make-up of the YTHDF1-knockout mouse faecal mycobiome was *Bacteroidetes* (accounting for 24.8%), while in WT mice, the faecal mycobiome was *Proteobacteria* (accounting for 24.8%). The third major phylum in the composition of the YTHDF1-knockout mouse faecal mycobiome was *Verrucomicrobia* (accounting for 15.7%), while in the WT mouse faecal mycobiome it was *Bacteroidetes* (accounting for 19.0%). At the order level, the results indicated many intriguing composition differences in the faecal mycobiome between YTHDF1-knockout and WT mice at 3 days post-CCI. The most abundant fungi in the YTHDF1-knockout group were *Eubacteriales* (26.5%), *Bacteroidales* (22.9%), *Lactobacillales* (16.0%), *Verrucomicrobiales* (15.7%), and *Erysipelotrichales* (7%), while the WT group mycobiomes were dominated by *Lactobacillales* (26.6%), *Eubacteriales* (23.5%), *Desulfovibrionales* (12.9%), *Bacteroidales* (9.0%), and *Chitinophagales* (9.0%).

For alpha diversity, the ACE and Chao1 indices show the abundance of the fungal mycobiome, while Shannon’s and Simpson’s indices indicate the diversity of the fungal mycobiome. The results demonstrated that the ACE and Chao1 indices of fungal microbiomes in YTHDF1-knockout mice were markedly higher than that in WT mice at 3 days after CCI (*p* < 0.05) (Figure 6A,B). However, the Shannon and Simpson indices of fungal microbiomes in YTHDF1-knockout and WT mice were similar (Figure 6C,D).

The beta diversity revealing the community distance between samples was evaluated by weighted and unweighted UniFrac distance (Figure 7A,B), which exhibited a significant difference (*p* < 0.05) in mycobiome profiles between the YTHDF1-knockout and WT groups at 3 days following CCI. The result of the clustering dendrogram of mouse faecal bacteria was consistent with the above conclusion (Figure 7C).

The differential taxa between YTHDF1-knockout and WT mice were analysed using the linear discriminant analysis effect size (LEfSe) method. The results in Figure 8A demonstrate the enriched fungal taxa in each group that had a >two-fold change and *p* < 0.05 (Kruskal-Wallis test). The faecal mycobiome of YTHDF1-knockout mice greatly diverged from that of WT mice at 3 days post-CCI (Figure 8B), revealing dissimilar mycobiome colonization. The fungi *Akkermansia*, *Verrucomicrobia*, *Muribaculum*, *Erysipelotrichia*, and *Dubosiella* were enriched in the YTHDF1-knockout group, while the fungi from various genera, such as *Proteobacteria*, *Kineothrix*, *Desulfovibrio*, *Chitinophagia*, and *Sediminibacterium*, were enriched in the WT mice.

### 2.4. YTHDF1-Knockout Affects Gene Expression in the Mouse Cerebral Cortex Post-TBI

To further investigate the influence of YTHDF1-knockout on gene expression, using the RNA sequencing data, we measured the levels of mRNA changes in mice cerebral cortex after TBI. Figure 9A reveals the principal component analyses of the YTHDF1-knockout and WT groups. It shows that the confidence ellipses of the samples among the YTHDF1-knockout and WT groups are separated from each other, suggesting that the gene expression patterns are similar within the same treatment group, but markedly different between the YTHDF1-knockout and WT groups (Figure 9A). The volcano plot (Figure 9B) reveals the significantly upregulated and downregulated mRNA in the YTHDF1-knockout group. We marked the top five most significantly upregulated genes (*Cdh15*, *Col23a1*, *Fcrls*, *Tox*, and *Slc17a6*) and top five downregulated genes (*Kdm5d*, *Ddx3y*, *Eif2s3y*, *Uty*, and *Lyrm7*) in the volcano plot. The heatmap reveals the relative expression levels of the three YTHDF1-knockout samples, and the three WT samples in the same group have similar presentation (Figure 9C). It shows that a total of 154 mRNAs had increased expression and 57 mRNAs decreased expression (*p* < 0.05, log2FC > 1) in YTHDF1-knockout mouse cerebral cortex after TBI (Figure 9D). The top 20 differentially changed mRNAs are displayed in Table 1.

Gene Ontology (GO) analyses revealed that the YTHDF1-knockout was associated with three parts of biological information—biological process: localization, aromatic amino acid family metabolic process, response to peptide hormone; cellular component: membrane, MKS complex, cytoplasmic microtubule; and molecular function: ionotropic glutamate receptor activity, glutamate receptor activity, GDP-dissociation inhibitor activity (Figure 10A). The Kyoto Encyclopedia of Genes and Genomes (KEGG) analyses showed the significantly changed genes in the YTHDF1-knockout mice. The most significant KEGG pathways related to these genes included neuroactive ligand-receptor interaction, glutamatergic synapse, axon guidance, and cAMP signalling pathway (Figure 10B).

## 3. Discussion

An increasing number of studies have focused on the role of RNA m^6^A methylation in the development of various neurological diseases, such as Parkinson’s disease [23], Alzheimer’s disease [24,25], multiple sclerosis [26], tumours [27,28,29], epilepsy [30], and neuropsychiatric disorders [31]. However, to date, there is limited research focusing on the role of RNA m^6^A methylation in TBI and BGA. We observed that YTHDF1-knockout could reduce the brain defect area, rescue neuron cells, and downregulate the levels of apoptosis, inflammation, and oedema proteins in the brain tissues of mice with CCI treatment. Meanwhile, YTHDF1-knockout increased the ratio of V/C and downregulated the levels of apoptosis and inflammation proteins in the gut tissues of mice post-CCI. Then, we identified that the DEGs in the cortex between YTHDF1-knockout and WT mice were primarily enriched in the regulation of neurotransmitter-related neuronal signalling pathways, inflammatory signalling pathways, and apoptotic signalling pathways. Our results revealed that the *ITGA6-*mediated cell adhesion molecules signalling pathway might be the key feature of m^6^A regulation in TBI-induced BGA dysfunction. Then, using RNA-Seq, we identified the DEGs in the cortex between YTHDF1-knockout and WT mice and investigated the possible m^6^A-mediated signalling pathways which were involved in TBI-induced BGA dysfunction.

After TBI, neuronal cells release a large number of pro-inflammatory cytokines, such as TNF-α, IL-1, and IL-6, leading to gut inflammation and damage to the gut structure. Our results showed that the deletion of YTHDF1 could mitigate structural lesions in both brain and gut tissues post-TBI. At the protein level, apoptosis is potentiated [32,33,34], marked by upregulated caspase3 [35,36,37,38], Cleaved caspase-3 [39] and BAX, as well as downregulated BCL2 [40,41]. Meanwhile, inflammation is enhanced [42], such as increased CD68 cells [42,43,44] and decreased Foxp3-mediated regulatory T cells (Tregs) [45,46,47]. In addition, brain oedema is marked by enhanced AQP4 expression [48] and attenuated claudin5 expression [49,50]. Our results indicated that the lack of YTHDF1 might block apoptosis, inflammation, and oedema following TBI, thereby attenuating TBI-induced BGA dysfunction in mice. 

GO and KEGG functional analyses showed that the DEGs in the mouse cortex post-TBI, in the YTHDF1-knockout group compared with the WT group, were primarily enriched in the regulation of neurotransmitter-related neuronal signalling pathways (neuroactive ligand-receptor interaction, glutamatergic synapse, axon guidance, and cAMP signalling pathway); response to inflammation (cell adhesion molecule signalling pathway); and the regulation of apoptotic process (calcium signalling pathway). These signalling pathways are involved in the pathophysiological processes post-TBI [51,52,53,54,55,56]. Among them, cell adhesion molecules play important roles in brain development, and maintaining synaptic structure, function, and plasticity [57,58]. An in vivo study demonstrated that TBI-induced intercellular adhesion molecule-1 (*ICAM-1*) regulated neuroinflammation and cell death through oxidative stress, vascular endothelial growth factor (VEGF), and matrix metalloproteinase (MMP) pathways. The deletion of *ICAM-1* exhibited a better outcome in alleviating neuroinflammation and cell death, as shown by markers such as Cleaved-caspase-3, IL-1β, NF-kB, and TNF-α. In the present study, we identified that integrin a6 (*ITGA6*) is one of the DEGs enriched in the cell adhesion molecule signalling pathway. *ITGA6* is commonly used as a glioblastoma stem-like cell (GSC) marker [59]. *ITGA6* inhibition can weaken the radioresistance of mesenchymal GSCs, and it decreases proliferation and stemness in proneural GSCs [60]. In addition, *ITGA6* also participates in the tumour development and progression of lower-grade gliomas [61]. However, the impact of *ITGA6* on TBI has not been reported to date. Our results provide a new perspective to further investigate the pathophysiological roles of *ITGA6*. Furthermore, m^6^A can affect mRNA stability [62], while the m^6^A reader YTHDF1 mainly promotes mRNA translation efficiency [17]. A clinical study revealed that in bladder cancer cells, m^6^A is highly enriched in the *ITGA6* transcripts, and increased m^6^A methylation of the *ITGA6* mRNA 3′UTR promotes the translation of *ITGA6* mRNA by binding YTHDF1 [63]. The deletion of YTHDF1 weakens the ability of RNA-binding proteins to identify m^6^A, thereby impeding mRNA translation and affecting downstream biological functions. Thus, YTHDF1 knockout likely downregulates the expression of *ITGA6* via the cell adhesion molecule signalling pathway, thereby alleviating neuroinflammation and cell apoptosis and ultimately attenuating TBI-induced BGA dysfunction. However, the detailed mechanisms need further investigation.

It is well known that the role of the gut microbiome in BGA exhibits a complex bidirectional communication system that includes neuroimmunoendocrine mediators and network pathways between the gut mucosa, ENS, and CNS [64]. The gut microbiome is a rich and complex ecosystem composed of bacteria, fungi, archaea, protists, viruses, and (sometimes) helminths [65]. While gut bacteria are essential for microbiome-BGA [66], mycobiome equilibrium is also critical for microbiome stability [67]. Mycobiome interactions may participate in mycobiome-BGA communication via immune- and nonimmune-regulated crosstalk systems, similar to those in the microbiome-BGA [68].

Increasing evidence indicates that TBI causes alteration of the gut microbiome by disrupting BGA [6]; meanwhile, host m^6^A modification affects the gut microbiome by inducing gut inflammatory responses [69,70]. M^6^A could act as a molecule to be involved in the interaction of the host and microbiome along with noncoding RNAs, chromatin remodelling, and histone modifications [15]. The knockout of YTHDF1 also significantly decreased gastric cancer cell proliferation and tumorigenesis in vivo [71]. Despite a relatively small number of gut fungi, they profoundly affect nutrition, metabolism, and immunity in the gut. Not only do gut fungi shape the functions of the gut, but they also affect the physiological functions of other crucial extraintestinal organs, such as the liver, lung, and brain [72]. For example, fungi are implicated in the inflammatory immune disorder of inflammatory bowel disease (IBD) [73,74,75], while mucosa-associated fungi (MAF) reinforced gut epithelial function and protected mice against gut injury and bacterial infection [76]. Meanwhile, fungi are able to synthesize and release neurotransmitters, which increases locomotor activity and aggressive behaviour and decreases anxiety reactions. Conversely, neuromediators may have an impact on gut fungi [65]. Therefore, we examined the alterations of fungal microbiome after TBI. In the current study, the results showed that the diversity of the mycobiome varied dramatically between YTHDF1-knockout and WT mice at 3 days following CCI. In alpha diversity, the ACE and Chao1 indices of the YTHDF1-knockout mice were markedly higher than those in WT mice, while Shannon’s and Simpson’s indices were similar. This indicated that YTHDF1 deficiency could promote the abundance of the fungal mycobiome but not the diversity of the fungal mycobiome after CCI. The beta diversity analysis showed the specific mycobiome structure detected from the group samples. The weighted and unweighted UniFrac distances indicated that the faecal mycobiomes of YTHDF1-knockout and WT mice had distinct community structures. The various characteristics of the mycobiome can be identified from family to species levels. The results of this study showed that the compositions of the mycobiome differed between the YTHDF1-knockout group and the WT group. *Akkermansia* exhibited a marked enrichment in YTHDF1-knockout mice. *Akkermansia muciniphilia* (*A. muciniphila*) belongs to *Akkermansia*, and it is regarded as a probiotic [77]. The abundance of *A. muciniphila* is positively associated with mucus layer thickness and it can protect gut barrier integrity in humans and animals [78,79,80,81]. Moreover, the colonization of A. muciniphila enhances the development of host innate and adaptive immune systems with anti-inflammatory effects [82]. The expressions of Foxp3 and retinoic orphan receptor gamma T (RORγt) in colonic tissue were both positively associated with *A. muciniphila* colonization, and *A. muciniphila* administration markedly promoted colonic RORγt^+^ Treg cell responses to ameliorate colitis [83]. *A. muciniphila* could promote 5-HT levels in the colons of mice via its outer membrane protein Amuc_1100 and *TLR2* signalling pathway, thus improving gastrointestinal diseases and metabolic disorders [84]. Additionally, a mouse experiment showed that *A. muciniphila* and its extracellular vesicles could enhance the 5-HT levels in the colon and hippocampus [85]. As the key mediator of the development and function of the ENS and CNS, 5-HT may play an important role in microbiome-BGA communication. 5-HT could improve certain gut bacterial growth and influence the gut microbiota via the host immune system, which likely affects the colonization and interaction of gut bacteria [84]. A correlation study indicated a high correlation between m^6^A genes and the *TLR2* gene [86]. *A. muciniphila* could influence specific m^6^A modifications in mono-associated mice [87]. In addition, YTHDF1 promotes the translation of m^6^A-modified mRNA [12,15]. YTHDF1-knockout weakens the ability of RNA-binding proteins to identify m^6^A, thereby impeding mRNA translation and affecting downstream biological functions. Thus, it is reasonable to deduce that the deletion of YTHDF1 may increase *A. muciniphila* colonization to enhance mouse anti-inflammatory effects, by promoting the expression of Foxp3 and promoting 5-HT levels, ultimately attenuating TBI-induced BGA dysfunction in mice.

Our study provides new therapeutic targets for BGA dysfunction after TBI. Attempts to develop agonists or inhibitors of these new molecular targets may offer a potential strategy to attenuate TBI-induced BGA dysfunction. Further studies are necessary to explore the mechanisms responsible for the participation of these genes in BGA dysfunction following TBI. This study harboured several limitations. First, we only elucidated DEGs correlated with TBI between the YTHDF1-knockout and WT mouse cortices, and whether the DEGs in the colon are consistent with those in the cortex post-TBI needs to be explored. Second, we only illustrated several possible pathways by which YTHDF1 regulates BGA dysfunction following TBI, and there may be more pathways that require further investigation. Third, we only described the DEGs in the cortex and the alteration of faecal mycobiomes between the YTHDF1-knockout and WT mice after TBI, and the mechanisms of YTHDF1 mediating TBI-induced BGA dysfunction remain to be explored in future studies. Fourth, our study only includes male mice; the data of female animals need to be further researched. In summary, this study revealed that YTHDF1 deficiency could attenuate TBI-induced BGA dysfunction, afforded a comprehensive analysis of the DEGs related to TBI, and further confirmed the hub genes associated with TBI progression.

## 4. Materials and Methods

### 4.1. Animals and Grouping

YTHDF1-knockout and C57BL/6J (WT) male mice, aged 8–12 weeks and weighing 24 ± 3 g, were provided by the Animal Center, Nanjing Agricultural University (Jiangsu, China). The YTHDF1-knockout mice were generated based on CRISPR/Cas9 [18]. sgRNA expression plasmids were generated by annealing and cloning oligos that were designed to target exon 4 of YTHDF1 into the BsaI sites of pUC57-sgRNA (Addgene 51132).

m YTHDF1-E4-1 T7 gRNA up: TAGGATAGTAACTGGACAGGTA

m YTHDF1-E4-1 gRNA down: AAACTACCTGTCCAGTTACTAT

m YTHDF1-E4-2 T7 gRNA up: TAGGCACCATGGTCCACTGCAG

m YTHDF1-E4-2 gRNA down: AAACCTGCAGTGGACCATGGTG.

The in vitro transcription and microinjection of CRISPR/Cas9 was performed as follows [88]: Briefly, the Cas9 expression construct pST1374-Cas9-N-NLS-Flag-linker-D10A (Addgene 51130) was linearized with Age I and transcribed using the mMACHINE™ T7 Ultra Kit (Ambion, AM1345). Cas9 mRNA was purified by an RNeasy Mini Kit (Qiagen, 74104). pUC57-sgRNA expression vectors were linearized by Dra I and transcribed using the MEGAshortscript Kit (Ambion, AM1354). sgRNAs were purified using the MEGAclear Kit (Ambion, AM1908). A mixture of Cas9 mRNA (20 ng/μL) and two sgRNAs (5 ng/μL each) was injected into the cytoplasm and the male pronucleus of zygotes obtained by the mating of CBF1. Injected zygotes were transferred into pseudopregnant CD1 female mice. Founder mice used for experiments were backcrossed to C57BL/6J for at least five generations. 

m YTHDF1-E4 C9 For: CACCTGAGTTCAGATCATTAC

m YTHDF1-E4 C9 Rev: GCTCCAGACTGTTCATCC.

Amplicon length: 650 bp. Applicable to genotyping founders and targeted embryonic stem cell (ESC).

The mice were housed in a standardized SPF animal laboratory under a 12 h light-dark cycle at a constant temperature (24 °C) and humidity (50%) and allowed free access to food and water. They were randomly assigned to 2 groups after 1 week of acclimation. (1) C57BL/6J + sham group (WT + sham, n = 3); (2) C57BL/6J + CCI group (WT + CCI, n = 9); (3) YTHDF1-knockout + sham group (YTHDF1-KO + sham, n = 3); (4) YTHDF1-knockout + CCI group (YTHDF1-KO + CCI, n = 9). There were no significant differences among the 4 groups in terms of general health, reactivity, locomotor activity, and neurological reflexes.

### 4.2. CCI Procedure

Sodium pentobarbital (65 mg/kg) was injected intraperitoneally to anaesthetize the mice before CCI injury, and surgery began when pedal reflexes were absent. A heating pad was used to maintain the core body temperature at 37 °C during surgery. The heads of mice were fixed in a stereotaxic frame, and a 4 mm diameter craniotomy was performed at 2.0 mm lateral to the midline over the right hemisphere and 2.0 mm posterior to bregma. A 3.0 mm rounded metal tip attached to the Pin-Point™ CCI device (Model PCI3000, Hatteras Instruments Inc., Cary, NC, USA) was angled vertically towards the brain surface. A severe injury was performed with 3.0 m/s speed, 2.0 mm depth, and 180 ms procedural duration. After the operation, the mice were removed from the stereotaxic holder, and the wound was lightly sutured. The sham groups underwent the same anaesthesia and surgical procedures but without CCI injury. All mice were placed in heated cages to maintain their body temperature after surgery, and the mice were not returned to their original cages until they were fully awake.

### 4.3. Sample Collection

In each of the 4 groups, 3 mice were used for haematoxylin-eosin staining (H&E) at 3 days post-CCI. The animals were euthanized by intraperitoneally injecting sodium pentobarbital (65 mg/kg), then perfused transcardially with phosphate-buffered saline followed by 50 mL of 4% paraformaldehyde. The brains were quickly dissected from the mouse body and fixed in 4% paraformaldehyde at 4 °C for 48 h. Coronal sections, which were obtained using a vibratome (Leica VT 1000S, Wetzlar, Germany), should contain the entire hippocampus (–0 mm, –3.5 mm relative to bregma). Serial coronal sections (30 μm thick) for H&E staining (n = 3 per group) were cut by a cryostat (Leica CM 1950). At the same time, the colon tissue was excised and fixed in a 4% paraformaldehyde solution, dehydrated using ethanol and xylene, and embedded in paraffin. Moreover, 5 mm thick sections were cut for H&E staining. In addition, the injured cerebral cortex (respective n = 3) and colon tissue (respective n = 3) were quickly removed, weighed, frozen in liquid nitrogen, and then stored at −80 °C for Western blotting, and the cerebral cortex on the injured side (respective n = 3) was used for RNA-Seq. Furthermore, the faecal samples (respective n = 3) were quickly collected, weighed, frozen in liquid nitrogen, and then stored at −80 °C for SMRT sequencing.

### 4.4. HE

Brain and colon sections were rinsed with dH_2_O, stained in haematoxylin for 6 min, and then decolorized in acid alcohol for 1 s. Afterwards, the sections were rinsed with dH_2_O for 3 s and counterstained in eosin for 15 s before being immersed in LiCO_3_. Next, the sections were rinsed with dH_2_O and dehydrated with 95% ethyl alcohol for 2–3 min and 100% ethyl alcohol for 2–3 min. Then, the sections were cleared with xylene for 2–5 min, mounted with DePeX (Thermo Fisher Scientific Inc., Waltham, MA, USA) in a fume hood, and visualized using an inverted microscope at 100× magnification (Nikon, Tokyo, Japan). Digital images were captured using a SPOT microscope camera (Diagnostic Instruments, Sterling Heights, MI, USA).

### 4.5. Western Blots

The total proteins from the brain and colon samples 3 days after CCI were extracted with ice-cold RIPA lysis buffer containing phosphatase inhibitors and protease inhibitors (Beyotime, Shanghai, China), and the supernatants were collected after the lysates were centrifuged at 4 °C. The supernatants were mixed with 5× dual colour protein loading buffer (Fudebio, Hangzhou, China) and boiled at 100 °C for 5 min. An equal amount of protein (30 µg/lane) was separated on an SDS-PAGE gel. The proteins were transferred onto a PVDF membrane. The membranes were developed with Femto ECL reagent (Fudebio, Hangzhou, China) after blocking and incubating with the primary antibodies and the secondary antibodies. The relative protein expression levels were analysed with GAPDH or β-tubulin as an internal control using Gel-Pro Analyser software (Media Cybernetics, Rockville, MD, USA). The primary antibody information is listed in Table 2.

### 4.6. PCR Amplification and SMRT Sequencing

Total DNA was extracted from the faecal samples using the E.Z.N.A.^®^ Soil DNA Kit (Omega Biotek, Norcross, GA, USA) according to the manufacturer’s protocols. The V1-V9 region of the bacterial 16S ribosomal RNA gene was amplified by PCR (95 °C for 2 min, followed by 27 cycles at 95 °C for 30 s, 55 °C for 30 s, and 72 °C for 60 s, with a final extension at 72 °C for 5 min) using primers 27F 5′-AGRGTTYGATYMTGGCTCAG-3′ and 1492R 5′-RGYTACCTTGTTACGACTT-3′, where the barcode is an eight-base sequence unique to each sample. PCRs were performed in triplicate in a 20 μL mixture containing 4 μL of 5× FastPfu Buffer, 2 μL of 2.5 mM dNTPs, 0.8 μL of each primer (5 μM), 0.4 μL of FastPfu Polymerase, and 10 ng of template DNA. Amplicons were extracted from 2% agarose gels and purified using the AxyPrep DNA Gel Extraction Kit (Axygen Biosciences, Union City, CA, USA) following the manufacturer’s instructions. SMRTbell libraries were prepared from the amplified DNA by blunt ligation according to the manufacturer’s instructions (Pacific Biosciences). Purified SMRTbell libraries from the Zymo and HMP mock communities were sequenced on dedicated PacBio Sequel II 8 M cells using Sequencing Kit 2.0 chemistry. Purified SMRTbell libraries from the pooled and barcoded samples were sequenced on a single PacBio Sequel II cell.

### 4.7. RNA-Seq

Total RNA from the cerebral cortices of WT and YTHDF1-knockout mice 3 days after CCI (each n = 3) was isolated using TRIzol reagent (Invitrogen, Waltham, MA, USA) following the manufacturer’s protocol. Afterwards, the quantity and quality of RNA were assessed using a NanoDrop ND-1000 (NanoDrop). RNA integrity was assessed using denaturing agarose gel electrophoresis. Next, mRNA was extracted using NEBNextR Poly (A) mRNA Magnetic Isolation Module (New England Biolabs, Hertfordshire, UK) according to the manufacturer’s procedure. Then, RNA libraries were prepared using a KAPA Stranded RNA-Seq Library Prep Kit (Illumina) following the manufacturer’s protocol. Finally, libraries were sequenced using Illumina HiSeq 4000 platforms.

### 4.8. Statistical Analyses

All data are presented as the mean ± standard error (SE). Student’s *t* tests were used to test the difference between two groups. A one-way ANOVA with Tukey’s multiple comparison test was used to evaluate the differences among multiple groups. The significance level was set to *p* < 0.05. All analyses were performed using SPSS version 25.0 (IBM, New York, NY, USA). GO and KEGG analyses of peaks and differentially expressed peaks were performed using R based on the hypergeometric distribution.

Bar charts show the most abundant phyla and species (>1%). Relative abundances were detected between sequencing technologies using paired Student’s t tests. The significant differences in taxa between the YTHDF1-knockout and WT groups at 3 days following CCI was compared with LEfSe, which uses a nonparametric factorial Kruskal–Wallis with a subsequent unpaired Wilcoxon test. An LDA higher than 4 and a *p* value lower than 0.05 were considered significant. Alpha diversity (ACE, Chao, Simpson, and Shannon indices) was compared between samples from the YTHDF1-knockout and WT mice by a t test with two dependent means and considering a *p* value < 0.05 as significant. Principle coordinate analysis (PCoA) plots and a clustering dendrogram were generated to visualize the beta diversity of the faecal mycobiome of the YTHDF1-knockout and WT groups.

## Figures and Tables

**Figure 1 ijms-24-04240-f001:**
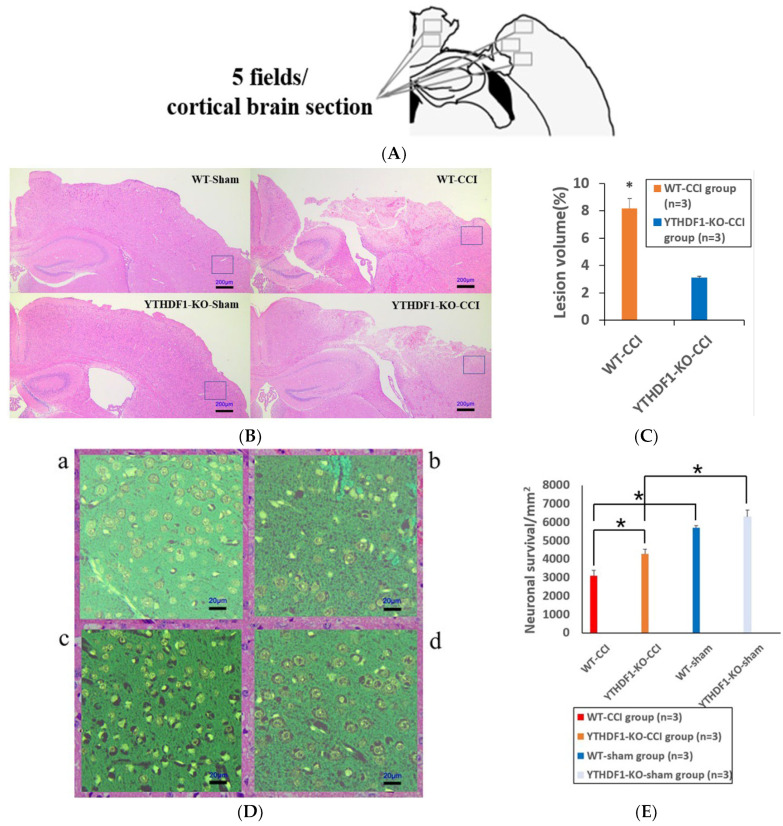
Histological images of HE coronal sections showing the brain tissue losses and neuronal survivals at 3 days post-CCI. (**A**) Diagrammatic representation of perilesional cortex sections at −1.20 mm from the bregma; 5 fields of the brain section were counted. (**B**) The lesion volume increased in CCI groups compared to sham groups (n = 3 mice per group). Scale bars: 200 μm. (**C**) The lesion volume was quantified (calculation formula: lesion volume (%) = (healthy side volume-injured side volume)/healthy side volume ×100%). * *p* < 0.05. (**D**) The neuronal survivals decreased in CCI groups compared to sham groups. (**a**) WT-sham group; (**b**) WT-CCI group; (**c**) YTHDF1-KO-sham group; (**d**) YTHDF1-KO-CCI group; Scale bars: 20 μm. (**E**) The number of neuronal survivals (5 fields/section) was quantified. Values are means ± SEs. * *p* < 0.05.

**Figure 2 ijms-24-04240-f002:**
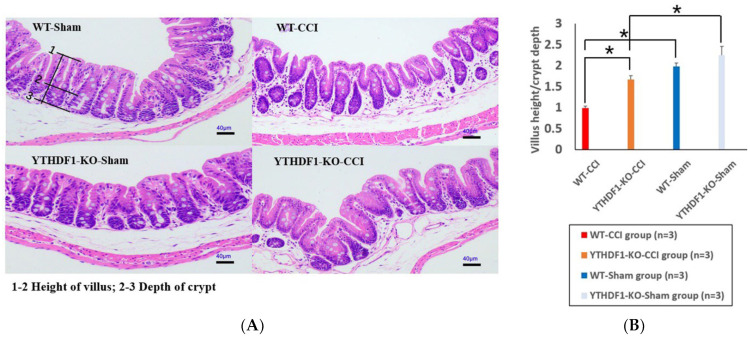
Histological images of HE colon sections showed that the V/C ratios of colon tissues decreased at 3 days post-CCI. (**A**) The villus height decreased and the crypt depth increased in CCI groups compared to sham groups. Scale bars: 40 μm. (**B**) The V/C ratios (10 fields/section) were quantified. Values are means ± SEs. * *p* < 0.05.

**Figure 3 ijms-24-04240-f003:**
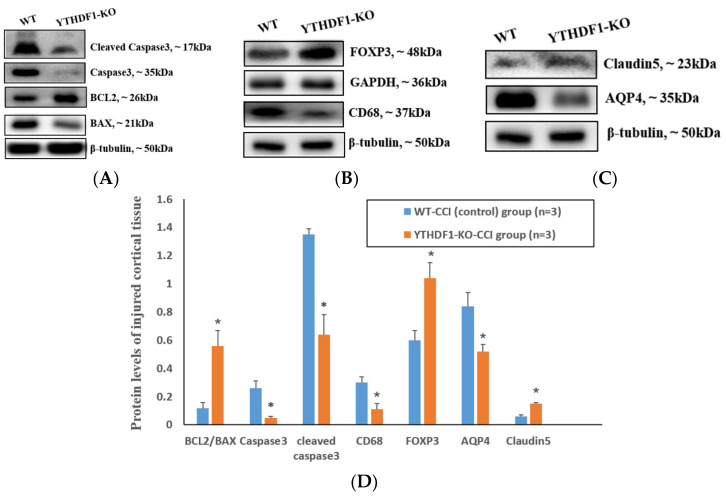
Western blots showed that YTHDF1-knockout markedly affected the apoptosis, inflammation, and oedema protein levels in injured cortical tissues at 3 days post-CCI. (BCL2, BAX, Caspase3, Cleaved caspase3, CD68, AQP4, and Claudin5 were normalized to β-tubulin, while FOXP3 was normalized to GAPDH). (**A**) The pro-apoptosis protein (BAX, Caspase3, and Cleaved caspase3) level was decreased, while the anti-apoptosis protein (BCL2) level was increased in the YTHDF1-knockout group compared to the WT group. (**B**) The pro-inflammation protein (CD68) level was markedly decreased and the anti-inflammation protein (FOXP3) level was significantly increased in the YTHDF1-knockout group compared to the WT group. (**C**) The pro-oedema protein (AQP4) level was markedly decreased and the anti-oedema protein (Claudin5) level notably increased in the YTHDF1-knockout group compared to the WT group. (**D**) Grayscale analyses of WB bands of BCL2/BAX, Caspase3, Cleaved caspase3, FOXP3, CD68, AQP4, and Claudin5 were quantified. Values are means ± SEs. * *p* < 0.05.

**Figure 4 ijms-24-04240-f004:**
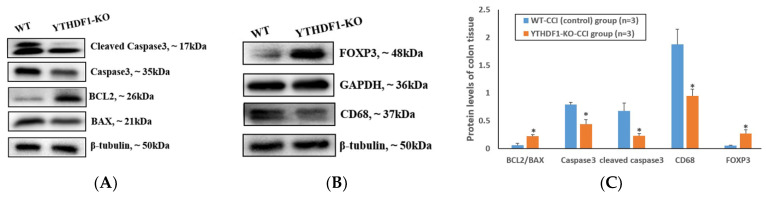
Western blots showed that YTHDF1 knockout significantly affected the apoptosis and inflammation protein levels in colon tissues at 3 days post-CCI. (BCL2, BAX, Caspase3, Cleaved caspase3, and CD68 were normalized to β-tubulin, while FOXP3 was normalized to GAPDH). (**A**) The pro-apoptosis protein (BAX, Caspase3, Cleaved caspase3) levels were decreased, while the anti-apoptosis protein (BCL2) level was increased in the YTHDF1-knockout group compared to the WT group. (**B**) The pro-inflammation protein (CD68) level was significantly decreased and the anti-inflammation protein (FOXP3) level notably increased in the YTHDF1-knockout group compared to the WT group. (**C**) Grayscale analyses of WB bands of BCL2/BAX, Caspase3, Cleaved caspase3, FOXP3, and CD68 were quantified. Values are means ± SEs. * *p* < 0.05.

**Figure 5 ijms-24-04240-f005:**
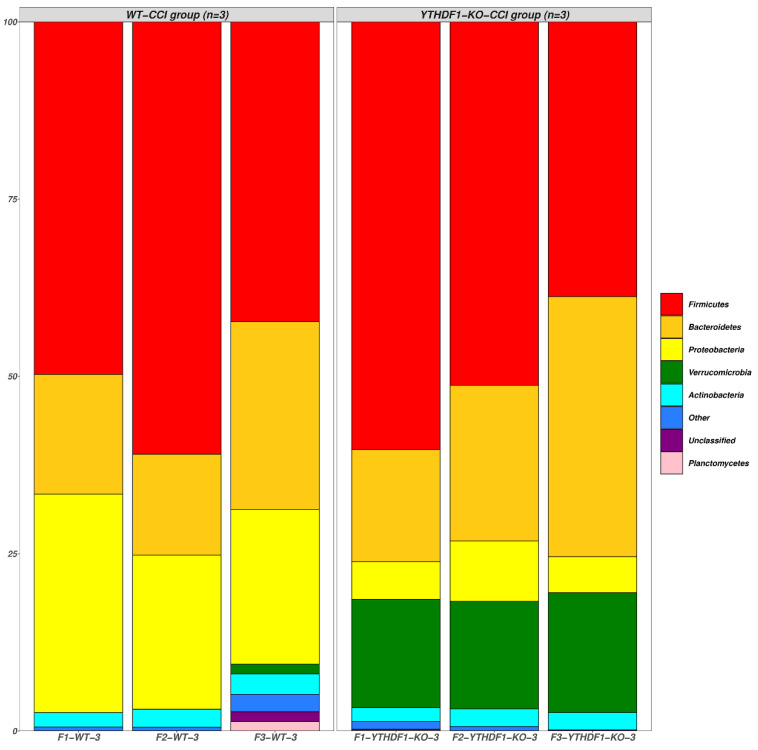
The bar plots show the relative phylum abundance of fungal microbiomes in faecal samples of WT mice and YTHDF1-knockout mice at 3 days post-CCI (YTHDF1-KO = YTHDF1-knockout). They were detected by high throughput sequencing on Internal transcribed spacer 2 (ITS2) of ribosomal DNA and analysed using the QIIME2 pipeline.

**Figure 6 ijms-24-04240-f006:**
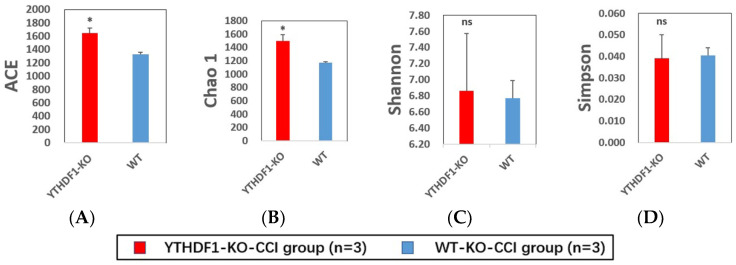
The comparison of alpha diversity of mycobiome in faecal samples of YTHDF1-knockout mice and WT mice at 3 days after CCI calculated by 4 different indices: (**A**) ACE index, (**B**) Chao1 index, (**C**) Shannon’s index, and (**D**) Simpson’s index. The bars indicate average diversity with the standard error of each group (YTHDF1-KO = YTHDF1-knockout) and statistically significant difference (* *p* < 0.05, ns = not significant).

**Figure 7 ijms-24-04240-f007:**
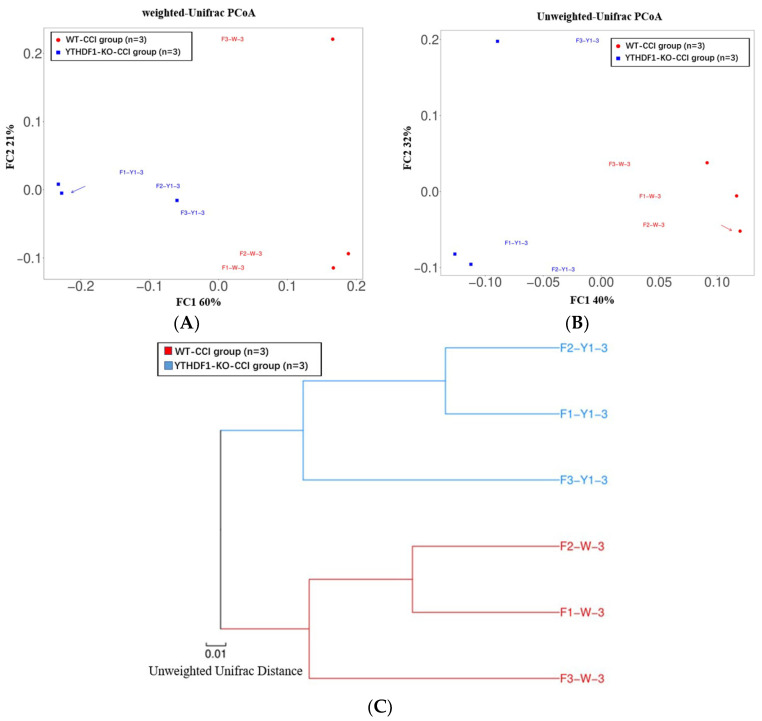
The beta diversity of mycobiome between samples was illustrated by principle coordinate analysis (PCoA) plots of (**A**) weighted and (**B**) unweighted UniFrac distance, showing significant differences of the mycobiome profile between YTHDF1-knockout and WT groups (tested by Permanova analysis with *p* < 0.05). (**C**) Clustering dendrogram of mice faecal bacteria using UPGMA (unweighted pair-group method with arithmetic mean).

**Figure 8 ijms-24-04240-f008:**
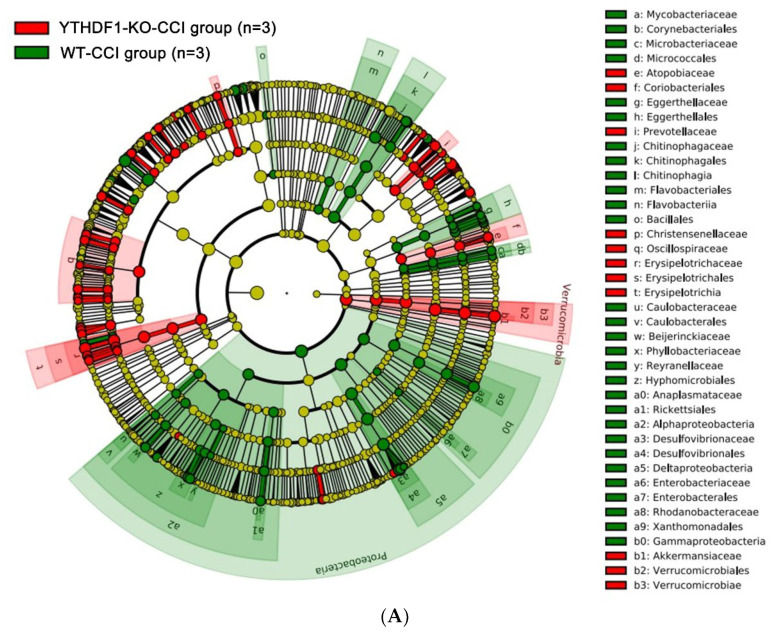
LEfSe analysis of faecal mycobiome of mice at 3 days following CCI reveals the markedly differential taxa between WT and YTHDF1-knockout (YTHDF1-KO) mice. (**A**) Cladogram (2-fold, *p* < 0.05). (**B**) LEfSe analysis for differential abundant taxa detected between WT and YTHDF1-knockout groups. Threshold parameters were set as *p* = 0.05 for the Mann–Whitney U test and multiclass analysis = all against all. Linear discriminant analysis (LDA) score > 4.0. (Green colour labels show the enriched fungal taxa in WT, while red labels show the taxa enriched in YTHDF1-knockout groups.

**Figure 9 ijms-24-04240-f009:**
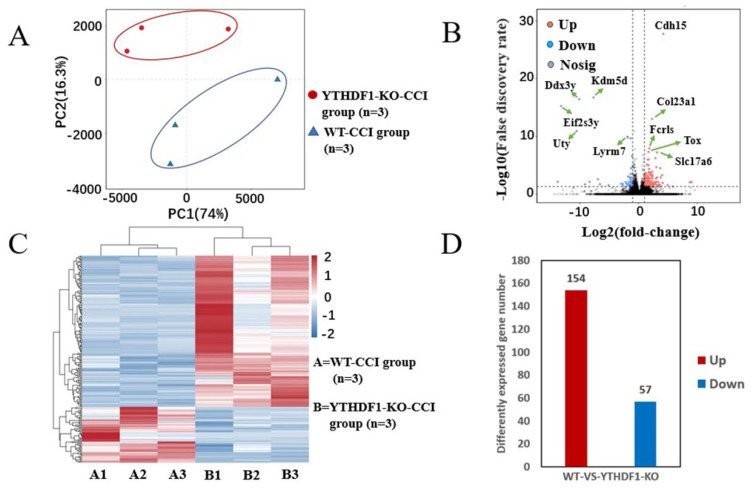
Different expression levels of mRNA between the YTHDF1-knockout group and the WT group after TBI. (**A**) Principal component analyses of the YTHDF1-knockout group and the WT group. The confidence ellipses of samples among YTHDF1-knockout and WT groups were separate from each other. (**B**) Volcano plot shows the markedly upregulated and downregulated mRNAs among YTHDF1-knockout and WT groups after TBI. (**C**) Heat map reveals the relative expression levels of the three YTHDF1-knockout samples and the three WT samples. (**D**) The expression levels of mRNA in YTHDF1-knockout mice. A total of 154 mRNAs had increased expression and 57 mRNAs decreased expression (*p* < 0.05, log_2_FC > 1) in YTHDF1-knockout mouse cerebral cortex after TBI.

**Figure 10 ijms-24-04240-f010:**
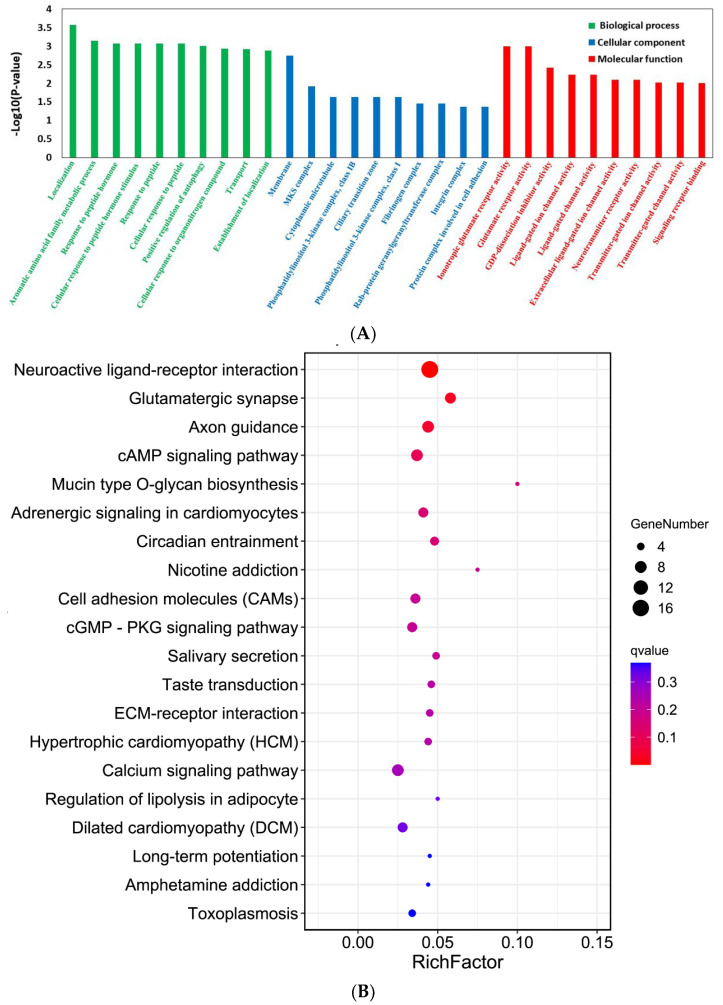
GO and KEGG analyses revealed the biological information of YTHDF1-knockout mice. (**A**) GO terms and (**B**) KEGG pathways most significantly related to YTHDF1-knockouts.

**Table 1 ijms-24-04240-t001:** The top 20 differently expressed mRNAs based on *p*-value.

mRNA	Lg (*p*-Value)	Log2 (Fold-Change)	Up/Down
Cdh15	−31.95	4.23	Up
Col23a1	−16.51	2.31	Up
Fcrls	−11.05	1.83	Up
Tox	−10.39	1.68	Up
Slc17a6	−10.27	3.15	Up
Qk	−9.76	0.71	Up
Tshz2	−9.17	2.87	Up
Dpy19l1	−8.88	0.91	Up
Myh7	−8.82	1.58	Up
Mid1	−8.82	1.69	Up
Kdm5d	−20.63	−7.65	Down
Ddx3y	−20.17	−10.10	Down
Eif2s3y	−18.85	−13.16	Down
Uty	−14.30	−10.55	Down
Plppr2	−14.26	−0.71	Down
Lyrm7	−13.23	−1.92	Down
Tmem114	−13.04	−1.83	Down
H2aw	−12.84	−1.34	Down
Grina	−10.90	−0.61	Down
Btg2	−9.58	−1.08	Down

**Table 2 ijms-24-04240-t002:** Antibody information.

Antibody	Manufacturer	Catalogue Number	Country	Dilution
BAX	Proteintech	60267-1-Ig	CN	1/20,000
Bcl2	Proteintech	26593-1-AP	CN	1/3000
Caspase-3	CST	9662s	USA	1/1000
Cleaved caspase-3	CST	9661s	USA	1/1000
FOXP3	Santa Cruz	sc-53876	USA	1/500
CD68	Abcam	ab283654	UK	1/1000
Claudin 5	Abcam	ab131259	UK	1/1000
Aquaporin 4	Abcam	ab128906	UK	1/1000
GAPDH	Proteintech	10494-1-AP	CN	1/20,000
β-tubulin	Abcam	ab131205	UK	1/5000

## Data Availability

Data are available on request.

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
