# Peer review of "YTHDF1 Attenuates TBI-Induced Brain-Gut Axis Dysfunction in Mice"

_ijms, 2023, doi:10.3390/ijms24044240_

Round 1

Reviewer 1 Report

This is a sound paper that describes interesting results that have the potential to contribute to advances in treatment of traumatic brain injury. The methods are sound and the manuscript is in good shape. Just a few things to consider:

1) Introduction – how does knocking out YTHDF1 interfere with N6-methyladenosine? Please summarize briefly for the reader.

2) Figure 10 – only the largest circle in the legend appears to correspond correctly to the circle sizes in the figure

3) For the most part the manuscript is written well but there are instances where grammar should be checked. For example:

”2.2. YTHDF1-knockout decreases pro-apoptosis, pro-inflammation and pro-edema proteins level 99 while increases the anti-apoptosis and anti-edema proteins level following CCI”

4) There would be no good reason for excluding females from this study and many reasons for including them (there are sex differences in both TBI outcomes and the gut-brain axis). A statement is needed that acknowledges both of those facts and that future studies need to also be performed in female animals. This could be included in the limitations paragraph at the end of the Discussion.

Reviewer 2 Report

This paper by Zhong and colleagues looks at the role of YTHDF1 in TBI with application to the brain gut axis. 

I have a number of significant problems with this paper 

1. There is no justification of why they look at YTHDF1, its picked at random with no reasoning. 

2. In the Western Blots figures they omit the control groups, this needs to be reversed/ 

3.  As with point 1, there is no reasoning or justification for looking at the fungal microbiome. Its just done and we are expected to understand why. 

4. The entire manuscript could do with an english language review. 

5. There are no n numbers in the figure legends. 

Round 2

Reviewer 2 Report

The authors have done a very good job of revising the paper, but I would prefer if points 1 and 3 from my rebuttal were included in the paper also. 
